# Transition Metal Catalyzed Hiyama Cross-Coupling: Recent Methodology Developments and Synthetic Applications

**DOI:** 10.3390/molecules27175654

**Published:** 2022-09-02

**Authors:** Rida Noor, Ameer Fawad Zahoor, Muhammad Irfan, Syed Makhdoom Hussain, Sajjad Ahmad, Ali Irfan, Katarzyna Kotwica-Mojzych, Mariusz Mojzych

**Affiliations:** 1Department of Chemistry, Government College University Faisalabad, Faisalabad 38000, Pakistan; 2Department of Pharmaceutics, Government College University Faisalabad, Faisalabad 38000, Pakistan; 3Department of Zoology, Government College University Faisalabad, Faisalabad 38000, Pakistan; 4Department of Chemistry, Faisalabad Campus, University of Engineering and Technology Lahore, Faisalabad 38000, Pakistan; 5Laboratory of Experimental Cytology, Medical University of Lublin, Radziwiłłowska 11, 20-080 Lublin, Poland; 6Department of Chemistry, Siedlce University of Natural Sciences and Humanities, 3-go Maja 54, 08-110 Siedlce, Poland

**Keywords:** Hiyama coupling, transition metals, organocatalysts, palladium nanoparticles, natural products

## Abstract

Hiyama cross-coupling is a versatile reaction in synthetic organic chemistry for the construction of carbon–carbon bonds. It involves the coupling of organosilicons with organic halides using transition metal catalysts in good yields and high enantioselectivities. In recent years, hectic progress has been made by researchers toward the synthesis of diversified natural products and pharmaceutical drugs using the Hiyama coupling reaction. This review emphasizes the recent synthetic developments and applications of Hiyama cross-coupling.

## 1. Introduction

Biaryl scaffolds are the prevailing structures that exist in numerous natural products [1,2,3], sensors [4], pharmaceuticals [5,6], ligands [7,8], polymers [9], agrochemicals [10], organocatalysts [11], fine chemical industries [12,13,14,15] and are scrutinized as the valuable intermediates in organic synthesis [16]. Several transition metal-catalyzed coupling protocols for the generation of the C-C bond have been reported in this regard, including Suzuki, Stille, Negishi, Kumada, and Hiyama coupling [17,18,19,20]. Among different protocols, Hiyama cross-coupling achieved remarkable attention by researchers for attaining biaryl moieties possessing a broad spectrum of pharmacological activities. For example, valsartan and losartan are used to treat high blood pressure and diabetic kidney disease. Felbinac is a non-steroidal anti-inflammatory drug (NSAID) used to cure muscular pain, inflammation, and sprains. Telmisartan is utilized as an alternative source for treating COVID-19 patients, and imatinib is a tyrosine kinase inhibitor (TKI) to inhibit cancer cell growth [21,22,23]. Hiyama coupling reveals to be an effective and convenient method of synthesizing stilbenoids. Stilbenoids have been found in medicinal plants and foods and exhibit potent biological activities such as antiviral, antifungal activity [24], anti-inflammatory property [25], neuroprotection [26], antioxidative property [27], and anticarcinogenic effect [28,29].

Aryl C-glucosides belong to a substantial class of natural compounds [30] and synthesized drugs [31,32]. Access to C-glycosides, which is an inhibitor of sodium–glucose cotransporter-2, via protecting a group-free Hiyama coupling reaction, has been developed. This strategy has been reported to synthesize pharmaceutically significant dapagliflozin compound that is useful to cure type 2 diabetes mellitus [33]. Hiyama cross-coupling offers 2-functionalized indoles formation by means of rhodium metal catalyst [34]. Furthermore, Hiyama coupling methodologies have been employed for the synthesis of diarylmethanes. Diarylmethane scaffolds containing natural products and drugs include segontin **1** (to cure the coronary heart disease) [35], benadryl **2** and tolpropamine **3** (antiallergic) [36,37], bifemelane **4** (antidepressant) [38] and piritrexim **5** (anticancer agent) [39]. Likewise, the diarylmethane containing marine natural product avrainvilleol **6** exhibits both antibacterial [40] and antioxidant [41] activities (Figure 1). 

Hiyama cross-coupling finds applications in synthesizing retinoids, benzofuran derivatives, benzoxocane, and picropodophyllin analogs, which impart a critical role in cell growth, embryo development, vision, immune response, and high affinity for tubulin [42,43,44,45,46,47,48].

Keeping the wide applications of the Hiyama coupling reaction in mind, here we have provided a comprehensive compilation of recent methodologies of the Hiyama coupling reaction.

## 2. Mechanistic Consideration

The mechanism of the Hiyama coupling reaction was proposed by Foubelo et al. in detail. The initial step of the mechanism involves oxidative addition of halide to Pd metal, resultantly converting palladium(0) to palladium(II). The transmetalation step leads to the splitting of the C-Si bond, and the cycle moves towards the construction of a new bond between carbon and palladium. In reductive elimination, the C-C bond is established, and a zero-valent state of palladium is achieved to restart a new cycle (Figure 1) [49]. 

## 3. Palladium-Catalyzed Hiyama Cross-Coupling

### Palladium Acetate as Catalyst

Palladium-catalyzed Hiyama cross-coupling reactions have extensively been employed for the synthesis of biaryl derivatives due to characteristic features of organosilane reagents such as non-toxicity, ease of access, high sustainability, and low cost [50,51,52]. Li and coworkers evaluated the catalytic efficacy of Pd(OAc)_2_/DABCO for the reaction of aryl halides and aryltrimethoxysilanes. This methodology covered a wide substrate scope by the addition of aryl bromides and iodides to aryl trialkoxysilanes giving biaryl derivatives in a moderate to excellent yield range (20–100%). Different solvents, including dioxane, MeCN, THF, acetone, and DMF, were investigated, and dioxane gave the highest yield. Out of the three different bases (KF, TBAF, and K_2_CO_3_), TBAF was selected as a suitable base for this reaction. A reference example is highlighted in Figure 2. *p*-Iodonitrobenzene **7** was coupled with phenyl trimethoxysilane **8**, affording the corresponding cross-coupled product **9** with quantitative yield. The coupling proceeded at 80 °C for 1 h using Pd(OAc)_2_ (3 mol%), DABCO (6 mol%), and tetrabutylammonium fluoride (TBAF) in dioxane [53].

A facile and effective protocol for the synthesis of symmetrically and asymmetrically functionalized (*E*)-1,2-diarylethenes by sequential one-pot Hiyama–Heck reactions was disclosed by Gordillo et al. The reaction proceeded in aqueous and ligand-free conditions. The symmetric (*E*)-1,2-diarylethene **12** was synthesized via one-pot Hiyama vinylation and Heck arylation of triethoxy(vinyl)silane **10** and aryl bromide **11** using Pd(OAc)_2_ (0.3 mol%) and NaOH in 98% yield with selectivities (100:0), respectively (Figure 3). For the synthesis of asymmetric (*E*)-1,2-diarylethene via one-pot Hiyama vinylation reaction, a different strategy was developed in which treatment of 3-bromopyridine **13** with triethoxy(vinyl)silane **10** by using Pd(OAc)_2_ aqueous NaOH and PEG followed by the Heck arylation in H_2_O afforded 100% conversion with 91% yield of corresponding asymmetric (*E*)-1,2-diarylethene **14** (Figure 4). Bromoarenes containing electron-donating and electron-withdrawing groups provided asymmetric (*E*)-1,2-diarylethenes in a moderate to excellent (71–91%) yield range [54].

The combination of palladium acetate with XPhos exhibits high efficacy in Hiyama cross-coupling of aryl mesylates and arylsilanes to generate corresponding biaryl derivatives in a 40–97% yield range. Wu and coworkers developed a general and efficient synthetic route for synthesizing the substituted biaryl compounds by palladium-catalyzed Hiyama cross-coupling reaction of unactivated aryl mesylates with arylsilanes. Various substituted aryl mesylates containing either electron-rich or electron-poor groups furnished the corresponding biaryl products in 40–97% yields. Maximum yield (97%) was attained in the case of coupling of methoxy substituted aryl mesylate **15** with triethoxy(phenyl)silane **16** using 4 mol% of Pd(OAc)_2_ as a catalyst, 10 mol% of XPhos **17** as ligand, and 2.0 equivalents of tetra-*n*-butylammonium fluoride (TBAF) as an additive in THF/*t-*BuOH mixture at 90 °C (Figure 5) [55].

Molander and Iannazzo outlined an impressive synthetic approach toward the synthesis of biaryls and heterobiaryl derivatives by Hiyama cross-coupling reactions of aryltrifluorosilanes with aryl and heteroaryl chlorides using a palladium catalyst. Aryl chlorides, bearing both electron-donating and withdrawing substituents, afforded corresponding derivatives in 70–98% yields. Phenyltrifluorosilane **19** was allowed to couple with methyl 3-chlorobenzoate **20** in the presence of 2.5 mol% palladium acetate and 5 mol% XPhos **17** using TBAF (2.5 equiv) as fluoride activator and *t-*BuOH as solvent. The reaction proceeded at 60 °C, providing the targeted product **21** in 98% yield (Figure 6). The coupling of a wide range of substituted heteroaryl chlorides with aryltrifluorosilanes provided the desired products in good to excellent yields (71–94%) [56].

Arenediazonium salts are the most reactive and efficient electrophiles for palladium-catalyzed synthetic organic chemistry [57,58,59,60,61,62]. Considering their importance, Qi and coworkers achieved the Hiyama coupling of dimethoxydiphenylsilane with mono or disubstituted arenediazonium tetrafluoroborate salts in a 65–89% yield range under mild reaction conditions. The dimethoxydiphenylsilane **22** was allowed to couple with bromo substituted arenediazonium tetrafluoroborate salt **23** using 5 mol% Pd(OAc)_2_ in methanol at room temperature for 6 h to obtain the targeted bromo substituted biaryl derivative **24** in 89% yield (Figure 7). The cross-coupling of 4-methylbenzenediazonium tetrafluoroborate salt with various organosilanes yielded the desired cross-coupling products in 78–87% yields [63].

Yuen et al. carried out a facile synthetic protocol for the formation of biaryl derivatives by Hiyama coupling reaction of aryl and heteroaryl chlorides with phenyl trimethoxysilane catalyzed by 0.2 mol% Pd(OAc)_2_/**26** to attain the cross-coupling products in moderate to good yield range (44–99%). The phenyl(trimethoxy)silane **8** was coupled with aryl chlorides **25** and **28** under H_2_O and solventless conditions, respectively, to obtain the desired products **27** and **29** in 97% and 99% yields, respectively. The reaction proceeded smoothly by using a 1:4 mixture of highly efficient Pd(OAc)_2_/**26** and TBAF·3H_2_O as the base at 110 °C under an N_2_ atmosphere (Figure 8). The aryl chlorides having electron-withdrawing groups (F, CF_3_, CO_2_Me, and CN) afforded a 47–99% yield range. Under solvent-free conditions, the reaction of heteroaryl chlorides and alkenyl chlorides with aryl trialkoxysilanes gave a 63–99% yield range, while in the case of H_2_O, a 36–97% yield range was attained. On the other hand, a 54–81% yield range was observed in the case of heteroaryl trialkoxysilanes [64].

Azetidines are distinctly comprised of four-membered azaheterocycle motifs due to possessing numerous chemical properties and ring strain [65,66]. Arylazetidine scaffolds are significant building modules and exhibit a diverse range of biological activities [67,68,69,70,71,72]. A convenient approach toward the synthesis of a variety of 3-arylazetidines derivatives through Hiyama coupling of 3-iodoazetidines with arylsilanes in mild reaction conditions was reported by Zou and coworkers. Triethoxy(phenyl)silane with electron-donating and -withdrawing substituents afforded the products moderate to excellent yields (30–88%). 1-Boc-3-iodoazetidine **30** was allowed to couple with 4-methylphenyltriethoxysilane **31** in the presence of Pd(OAc)_2_ catalyst (5 mol%), phosphine ligand (10 mol%) **32**, tetra-*n*-butylammonium fluoride (TBAF in THF) and dioxane under argon atmosphere to obtain the targeted product **33** in an excellent (88%) yield along with **34** in 99:1 ratio, respectively (Figure 9). The substrate scope of heterocycloalkyl iodides was observed under optimized conditions. The desired coupling products were obtained in a 33–85% yield range [73]. 

Dihetaryl disulfides possessing pyrimidine rings exhibit diversified biological and pharmacological activities [74,75,76], such as antifungal, antibacterial, and calcium-channel modulation [77]. An interesting and modular strategy for the formation of carbon–carbon bonds to produce potent biologically active compounds through palladium-catalyzed copper-promoted Hiyama cross-coupling reaction has been reported by Liu et al. Dihetaryl disulfides containing electron-rich and electron-poor substituents in the *para*-positions of benzene rings afforded corresponding products in moderate yield range. A maximum yield (78%) of **36** was observed in the case of cross-coupling of dihetaryl disulfide **35** with trimethoxy(phenyl)silane **8** using 3 mol% Pd(OAc)_2_ catalyst, an efficient activator copper (I) thiophene-2-carboxylate (CuTC), TBAF and 6 mol% PCy_3_ ligand using THF as the only solvent (Figure 10) [78].

The palladium NNC-pincer complex was found to be an efficient catalyst precursor for the generation of monomeric palladium(0) species. The applications of palladium NNC-pincer complex as a catalyst in the allylic arylation of allyl acetates with sodium tetraarylboronates [79] and in the Heck reaction of aryl halides with activated alkenes [80] gained tremendous importance in recent years. Keeping its efficiency under consideration, Ichii et al. synthesized corresponding biaryls via palladium NNC-pincer complex catalyzed Hiyama coupling reactions of aryl bromides with aryl(trialkoxy)silanes. Bromobenzenes having electron-donating and electron-withdrawing substituents resulted in a moderate to excellent yield range (39–99%) of biaryl products. The substituted aryl bromide **11** was allowed to couple with phenyl trimethoxysilane **8** using 5 mol ppm of palladium complex **37**. KF was selected as a suitable base for the corresponding reaction. The maximum yield (99%) of biaryl derivative **38** was observed using propylene glycol as an effective solvent (Figure 11) [81]. 

Amide is the versatile and valuable functionality widely utilized as a building block in biologically important compounds such as proteins and enzymes due to the sturdy nature of the amide C-N bond [82,83]. Highlighting the significance of the amide C-N bond, Idris and Lee devised a route for the Pd-catalyzed transformation of amides to corresponding aryl ketones. A number of different substituted *N*-acylglutarimides were reacted with phenyl triethoxysilanes to give targeted products in (0–98%) yield. The highest yield (98%) was observed by coupling of *N*-4-fluorobenzoylglutarimide **39** and phenyltriethoxysilane **16** using palladium acetate (2 mol%), PCy_3_ (4 mol%) in 1,4-dioxane/H_2_O, Et_3_N**^.^**3HF and LiOAc at 90 °C for 6 h (Figure 12). *N*-Acylglutarimide bearing sterically bulky *tert*-butyl group provided no desired coupling product. The coupling of substituted *N*-acylglutarimides with a broad range of arylsiloxanes afforded corresponding coupled products up to 93% yield [84].

The trapping of σ-alkylpalladium intermediate using arylsilanes under palladium-catalyzed Domino Heck/Hiyama coupling was disclosed by Wu and coworkers. This approach demonstrated the broad substrate scope and compatibility with various functional groups. Different aryl-tethered activated or unactivated alkenes were treated with substituted arylsilanes by means of palladium acetate catalyst in MeCN at 80 °C under argon atmospheric conditions. Consequently, the corresponding products were obtained in 62–88% and 53–81% yields, respectively. Excellent results were observed in the case of coupling of *N*-(2-iodo-4-methoxyphenyl)-*N*-methylmethacrylamide **41** and methoxy substituted phenyl triethoxysilane **42** using an effective ligand PPh_3_ (10 mol%) and Bu_4_NF giving 88% yield to furnish the respective product **43** (Figure 13). This methodology was employed to yield the ezetimibe, a cholesterol absorption inhibitor, analogs using *N*-(2-iodophenyl)-*N*-methylmethacrylamide and ezetimibe-derived arylsilane [85]. 

The azaindoline framework is a nitrogen-bearing heterocycle that exists in several complex compounds and exhibits numerous biological and pharmaceutical attributes [86]. Keeping these characteristics in mind, Ye et al. disclosed the formation of azaindoline derivatives, which was carried out by the Domino Heck cyclization/Hiyama coupling protocol. A number of functionalized azaindoline derivatives were achieved in 46–85% yields. A series of ligands (PPh_3_, P(2-furyl)_3_, XPhos, SPhos, P(4-MeOC_6_H_4_)_3_ were screened, and the electron-rich P(4-MeOC_6_H_4_)_3_ was declared as a suitable ligand for the corresponding reaction. The reaction of Bn-protected aminopyridine **44** and 4-methoxy substituted phenyltriethoxysilane **42** was progressed in 1,4-dioxane using 5 mol% Pd(OAc)_2_ catalyst and 2.0 equivalents of Bu_4_NF. The targeted coupling product **45** was furnished in 85% yield by maintaining the temperature at 80 °C (Figure 14). Excellent results (46–85%) were obtained in the case of electron-donating (phenyl, OMe, CH_3_) and -withdrawing (F, CF_3_, Cl) substituents. Moreover, heteroarylsilanes, including 2-thienylsilane and 3-thienylsilane, provided access to targeted products in 77% and 82% yields, respectively [87].

## 4. Palladium Chloride as Catalyst

Clark, in 2005, reported the Hiyama cross-coupling under microwave irradiation for the first time. The microwave-assisted reaction of 3-bromotoluene **46** was accomplished with phenyl trimethoxysilane **8** in the presence of 1.25% [Pd(allyl)Cl]_2_, ligand **47**/*N*-mepip solution and tetra n-butylammonium fluoride (TBAF) promoter (1 M soln. in THF) to achieve the targeted coupling product **48** in >95% yield with >99% conversion (Figure 15). A fascinating application of Hiyama coupling involves the synthesis of arylalkenes. In this regard, *para*-chloroacetophenone **49** underwent cross-coupling with vinyltrimethoxysiloxane **50** utilizing catalyst derived from 1.25% [Pd(allyl)Cl_2_] and ligand **47**/*N*-mepip solution using TBAF in THF at 110 °C under microwave irradiation to afford colorless styrene derivative **51** in 95% yield (Figure 16) [88]. 

Organosilanes have attracted the attention of research groups due to their assorted benefits (such as low cost, ease of availability, nontoxic byproducts, and stability) as compared to the other organometallic precursors (organoboron, organostannane, organozinc, organotin). Unsuccessful attempts to couple 2-trimethylsilylpyridine with 4-iodoanisole diverted the attention of researchers towards sustainable liquids, i.e., chloropyridyltrimethylsilanes for Hiyama cross-coupling reaction. Chloropyridyltrimethylsilanes could be afforded either by halogen or hydrogen lithium exchange of chloropyridines followed by the reaction with chlorotrimethylsilane. Pierrat et al., in 2005, disclosed the Hiyama cross-coupling of chloro-substituted pyridyltrimethylsilanes with aryl halides. 1-Fluoro-4-iodobenzene **52** was allowed to react with chloro substituted pyridyltrimethylsilanes **53** using 5% PdCl_2_(PPh_3_)_2_, 10% PPh_3_, and copper iodide (1 equiv) to carry out the coupling in DMF solvent. The reaction was conducted using two equivalents of tetra-*n*-butylammonium fluoride (TBAF in THF) as a suitable base at room temperature for 12 h, affording the targeted biaryl product **54** in 95% yield (Figure 17) [89]. 

The phosphine-free palladium-catalyzed synthesis of *para*-substituted biaryl scaffold using air-insensitive PdCl_2_/hydrazone ligand was reported by Mino and coworkers. Several functionalized substituted biaryls were obtained in a 50–90% yield range via the reaction of aryl bromides with different siloxanes. An excellent result (90%) was attained by the reaction of phenyl bromide **55** with *para*-substituted phenyl triethoxysilane **56** in the presence of air-stable phosphine-free PdCl_2_/hydrazone ligand **57** using TBAF in THF under argon atmosphere. After screening a variety of solvents (DMSO, dioxane, *t-*BuOH, toluene, and DMF), toluene was selected as a suitable solvent for the corresponding reaction (Figure 18) [90].

Multicomponent assembly protocol fascinates synthetic chemists due to its efficacy in the synthesis of molecular complexity and is acceptable for diversity-oriented synthesis [91,92]. An efficient approach for the synthesis of multisubstituted unsymmetrical biaryls was reported by Akai and coworkers in 2008. The reaction of substituted 8-TBDMS-1-naphthols with a wide range of aryl iodides afforded the biaryl derivatives moderate to excellent yields (46–81%). The fluoride-free Hiyama coupling of **59** was accomplished with 4-cyanoiodobenzene **60** giving a multisubstituted asymmetrical biaryl through the formation of pentacoordinate silicate **61** as an intermediate. [(allyl)PdCl]_2_, AsPh_3_, Cs_2_CO_3_, and dimethoxyethane (DME) were selected as an appropriate catalyst, ligand, base, and solvent, respectively, to afford the maximum yield (81%) of targeted asymmetrical biaryl derivative **62** under argon atmosphere (Figure 19) [93]. 

Pincer-type palladium complexes are intriguing due to their promising reactivity and stability. Inés et al. synthesized the non-symmetric PCN pincer-type palladium complex by the reduction of 1-(3-nitrophenyl)pyrazole to amine, which upon treatment with ClPPh_2_ provided an unreliable and air-sensitive ligand. The ligand further reacted with Pd(COD)Cl_2_ to yield a targeted non-symmetric pincer-type complex that efficiently catalyzed the Hiyama cross-coupling reaction in eco-friendly reaction media. Excellent results were achieved using two methods in the presence of an efficient catalyst. The first pathway was concerned with the reaction of phenyl trimethoxysilane **8** with **11** catalyzed by 2 mol% **63** using NaOH in H_2_O at 140 °C for 3 h to obtain the corresponding biaryl derivative **38** in 82% yield. The alternative developed method involved the coupling of phenyl trimethoxysilane **8** with **11** in the presence of 4 mol% of catalyst **63** using *n*-Bu_4_NF and *o*-xylene at 80 °C for 4 h to afford the required biaryl product **38** in 61% yield (Figure 20) [94].

Chen et al. synthesized the sustainable cationic bipyridyl ligand by reacting 4,4^′^-bis(bromomethyl)-2,2′-bipyridine with 50% aqueous solution of trimethylamine in CH_2_Cl_2_ that catalyzed the Hiyama cross-coupling reaction efficiently. 4-Bromoanisole **64** was reacted with phenyl triethoxysiloxane **16** in H_2_O using Pd(NH_3_)_2_Cl_2_/**65** (0.1 mol%) as a catalyst. NaOH was screened as a suitable base to attain the desired biaryl derivative **66** in 99% yield (Figure 21). The Hiyama cross-coupling of several aryl bromides with a variety of triethoxy(aryl)silanes gave substituted biaryl derivatives in (35–99%) yield range [95].

Aryl imidazol-1-ylsulfonates are the electron-deficient substrates that promote the palladium-catalyzed Hiyama and Sonogashira cross-coupling under copper-free conditions, as reported by Williams and coworkers. Several functionalized biaryl derivatives were obtained by the reaction of aryl imidazylates with (2-hydroxymethylphenyl) dimethyl silane (HOMSi) reagent in 72–99% yields. The 2-naphthyl imidazol-1-ylsulfonate **67** was coupled with 4-methoxyphenyl HOMSi reagent **68** in the presence of Pd(dppf)Cl_2_ (0.05 equiv) in dry DMSO with K_2_CO_3_ at 65 °C giving the cross-coupling product **69** in 99% yield (Figure 22) [96]. 

A research group by Hughes also reported a facile and convenient methodology for the palladium-catalyzed synthesis of biphenyls in 2011 using the HOMSi^®^ reagent. In this methodology, the aryl bromides and iodides were proved to be more efficient substrates to couple with the HOMSi^®^ reagent in DMF solvent. Excellent (>98%) conversion was achieved by the reaction of HOMSi^®^ reagent **71** with **72** using PdCl_2_ as the catalyst, **73** as ligand using CuI, and K_2_CO_3_ as the base at 80 °C for 15 h. DMF was selected as the optimal solvent by observing the results with THF, MeOH, 1,4-dioxane, and DMSO (Figure 23). Compatibility with functional groups, reactions under fluoride-free conditions, and recycling of organosilicon byproducts are the salient features of the HOMSi^®^ reagent [97]. 

Stilbenes and their hydroxylated derivatives are of tremendous interest as they exhibit diversified biological activities such as fungistatic, antibacterial and anticancer [98,99,100,101]. Due to their extended conjugation system, stilbenes have been utilized to assemble electronic and optoelectrical devices, i.e., solar cells, LEDs, and dye lasers [102,103]. Resveratrol and combretastatin A-4 are a few bioactive derivatives of stilbene that can be obtained naturally from the extraction of plants. An effective catalytic system was established for the stereoselective formation of substituted *E*-stilbenes using Heck cross-coupling reaction while studying the catalytic activity of Pd complexes with H-spirophosphane ligands. Hiyama coupling reaction emerged to be a more effective route to synthesize substituted *E*-stilbenes. Keeping these considerations in mind, Skarżyńska and coworkers, in 2011, proposed an efficient synthesis of *E*-stilbenes via Hiyama coupling reactions of substituted styrylsilanes with aryl halides catalyzed by an efficient and stereoselective precatalyst. The styrylsilanes possessing either electron-deficient groups or *para*-substituted electron-rich groups resulted in an excellent yield range (87–99%). A 100% conversion of iodobenzene and 99% yield of bromo substituted *E*-stilbene **77** was obtained by the addition of **75** to iodobenzene **76** using palladium complex with *H*-spirophosphorane ligand [PdCl_2_P(OCH_2_Cme_2_NH)OCH_2_Cme_2_NH_2_] as precatalyst in the presence of [*^n^*Bu_4_N]F additive. The reaction progressed in tetrahydrofuran solvent at 60 °C for 12 h (Figure 24) [104].

Cheng et al. disclosed a PdCl_2_-catalyzed Hiyama-type coupling of arenesulfinates and organosilanes in mild reaction conditions. A wide range of substituted biaryls was obtained in 73–94% yield by coupling various arenesulfinates with aryl trialkoxysilanes. The coupling of *p*-methylbenzenesulfinate **78** with phenyl triethoxysilane **16** was carried out in the presence of PdCl_2_ (5 mol%) and TBAF (additive) in the THF solvent. The reaction progressed well under aerobic conditions at 70 °C giving the corresponding biaryl product **79** in 94% yield (Figure 25) [105]. 

Chang et al. reported a facile protocol for synthesizing the biaryl ketones via carbonylative Hiyama coupling reaction. In this regard, the phenyl triethoxysilane **16** was allowed to react with 4-bromophenyl iodide **80** using PdCl_2_(MeCN)_2_ catalyst and cesium fluoride (CsF) promoter to afford the corresponding ketone **81** in 94% yield (Figure 26). The reaction temperature was maintained at 80 °C. Different solvents (DMSO, toluene, DMF, CH_3_CN, and 1,4-dioxane) were screened, and NMP (*N*-methyl-2-pyrrolidone) was selected as the optimal solvent for the corresponding reaction. Aryl halides having electron-withdrawing (-Cl, -F, -NO_2_, CN) and electron-donating substituent (-CH_3_) gave moderate to excellent yields (68–94%) [106].

Diarylmethanes have attained huge interest as they exist in biologically active natural compounds and drugs [107,108]. For example, avrainvilleol, a marine natural product, possesses a diarylmethane motif and exhibits antibacterial [40] and antioxidant [41] activities. Several drugs containing diarylmethanes demonstrate a wide range of biological activities such as segontin (used to cure coronary heart disease) [35], bifemelane, tolpropamine, and piritrexim act as an antidepressant, antiallergic and anticancer agents, respectively [36,37,38,39]. Functionalized diarylmethane derivatives were achieved by coupling a variety of benzylic ammonium salts with phenyl trimethoxysilane, as elaborated by Zhao and coworkers in 2019. Methoxy substituted benzyltrimethylammonium salt **82** was treated with phenyl trimethoxysilane **8** to acquire the desired diarylmethane derivative **83** in 92% yield. The reaction proceeded well at 120 °C in the presence of 5 mol% PdCl_2_(CH_3_CN)_2_ catalyst, 20 mol% PPh_2_Cy ligands, and TBAF in EtOH (Figure 27). In the case of aryl trimethoxysilanes, the substrate containing electron-donating groups afforded corresponding diarylmethane derivatives in (57–97%) yield range. Moreover, the substrate-bearing heterocycles, including thiophene and furan, gave coupling products a 94% yield [109].

Synthesis of vinylsilanes is a very promising task in research due to their spacious applications in organic syntheses, such as the Hiyama coupling reaction [110]. Wisthoff et al. designed a facile protocol to accomplish the synthesis of *cis*- or *trans*-tetrasubstituted vinylsilanes via carbosilylation of three components, including symmetrical alkynes, alkyl zinc iodides, and iodosilanes using a palladium catalyst. The authors also reported conditions for tetrasubstituted vinylsilanes via the Hiyama coupling reaction that facilitated the synthesis of tetrasubstituted alkenes. The *cis*- or *trans*-tetrasubstituted vinylsilanes were synthesized in a 32–97% yield range by the addition of iodosilane to the solution of alkyl zinc iodide and alkyne using (Ph_3_P)_2_PdCl_2_ (2 mol%) catalyst in dioxane. Triethylamine (Et_3_N) was reported as a suitable base for this reaction. Tetrasubstituted vinylsilanes underwent Hiyama cross-coupling reactions to achieve the stereodefined tetrasubstituted alkenes. The reaction of **84** with 1-bromo-3-methybenzene **46** was conducted in the presence of KOSiMe_3_, and 18-crown-6 in THF at 65 °C, followed by the treatment with 2.5 mol% [(allyl)PdCl_2_] catalyst and 5 mol% SPhos ligands at 65 °C afforded the desired geometrically defined tetrasubstituted alkene **85** in 62% yield (Figure 28). The effect of ligands on the selectivity of alkyl-substituted tetrasubstituted vinylsilanes was also observed. It was found that the use of DrewPhos ligand **86** (Figure 2) gave results in (54–92%) yield of alkyl-substituted tetrasubstituted vinylsilanes with excellent *syn*-selectivity while using JessePhos **87** (Figure 2) as ligand resulted in 61–92% yield range of alkyl-substituted tetrasubstituted vinylsilanes with excellent *anti*-selectivity [111]. 

γ-Valerolactone-based tetrabutylphosphonium 4-ethoxyvalerate are effective and versatile biomass-derived ionic liquids and have been extensively utilized in synthesis protocols due to their tunability. In order to elaborate on the effect of ionic liquid, Orha et al. outlined the efficient synthesis of substituted biaryl derivatives by the addition of iodoaromatic substrates to triethoxyphenylsilane. In this methodology, reactions of several aryl iodides were performed with triethoxyphenylsilane to give numerous substituted biaryl structures in good yields (45–72%). 1-Iodo-4-methylbenzene **88** was coupled with triethoxyphenylsilane **16** in tetrabutylphosphonium 4-ethoxyvalerate [TBA][4EtOV] afforded the desired biaryl derivative **79** in 72% yield. The reaction was carried out at 130 °C using Pd(PPh_3_)_2_Cl_2_ as catalyst precursor and a fluoride source (TBAF) to activate the transmetalation reagent (Figure 29) [112].

## 5. Pd/C as Catalyst

A heterogeneous catalytic system proved to be an effective system over a homogeneous catalytic system in the organic synthetic field regarding their stability, recoverability, and ease of handling [113]. Considering the efficacy of heterogeneous catalysts, Yanase et al., in 2011, reported the synthesis of biaryl derivatives via water-mediated Hiyama cross-coupling reaction using Pd/C, a heterogeneous catalyst [114] with electron-deficient phosphine ligand [(4-FC_6_H_4_)_3_P]. Several functionalized biaryl derivatives were attained in a 47–90% yield range by cross-coupling of aryl halides with aryltrialkoxysilanes. The reaction was progressed at 120 °C using 0.5 mol% of 10% Pd/C catalyst, 1 mol% of tris(4-fluorophenyl)phosphine as ligand, TBAF**^.^**3H_2_O as an activator, and 4.8% aqueous toluene. A 90% yield of targeted biaryl product **18** was obtained by coupling 3-methoxyphenylbromide **89** with phenyl triethoxysilane **16** (Figure 30) [115]. Later on, in a 2013 research group by Monguchi, they developed a facile and efficient methodology to synthesize the biaryl derivatives in (47–90%) yield range using (4-C_6_H_4_)_3_P (1 mol%) ligand [116]. 

The very first ligand-free Pd/C catalyzed Hiyama cross-coupling reaction for the construction of a variety of biphenyl derivatives was developed by Sajiki and coworkers in 2012. A series of solvents (DMF, THF, CH_3_CN, EtOH, toluene) and fluoride sources (TBAF, LiF, CsF, KF) were screened, and among them, toluene and TBAF·3H_2_O were selected as a suitable solvent, and fluoride source for corresponding Pd/C catalyzed Hiyama cross-coupling reaction. Aryl bromide **11** was reacted with methoxy substituted phenyl triethoxysilane **42** to afford a maximum (90%) yield of desired biphenyl derivative **90**. The reaction was refluxed for 24 h in toluene containing 0.5 mol% of 5% Pd/C catalyst, TBAF·3H_2_O as an activator, and acetic acid (Figure 31) [117]. 

γ-Valerolactone (GVL), a safe and sustainable bio-based chemical obtained from lignocellulosic biomass [118], was found to be an efficient media for Hiyama coupling reactions. The use of highly effective and non-expensive heterogeneous catalyst Pd/C with γ-valerolactone competently promotes the eco-friendly Hiyama cross-coupling reaction under practical and mild conditions without using any ligand or additive as reported by Ismalaj et al. A number of aryl halides were treated with phenyl triethoxysilanes using 1 M GVL as an efficient biomass-based solvent for Hiyama coupling reaction. Tetrabutylammonium fluoride (TBAF) was found to be the best activator of organosilanes, while CsF and KF were not such efficient activators. An excellent result (94%) of desired biaryl product **92** was obtained by carrying out the reaction of bromobenzene **55** with phenyl triethoxysilane **16** using Pd/C (0.5 mol%) as catalyst in 1M GVL **91** solvent and TBAF for 24 h at 130 °C (Figure 32) [119].

## 6. Pd/Fe_3_O_4_ as Catalyst

An efficient methodology for the synthesis of a diverse range of biaryl derivatives via Hiyama cross-coupling reaction of aryl bromides with aryl siloxanes using a highly efficient, easily recoverable, and eco-friendly Pd/Fe_3_O_4_ catalytic system was proposed by Sreedhar and coworkers. Fe_3_O_4_ supported palladium catalyst shows superparamagnetic behavior, and its catalytic activity remains unchanged after recycling five times. The mechanistic studies showed that Pd/Fe_3_O_4_ catalyzed Hiyama cross-coupling reaction proceeds through oxidative addition and transmetalation followed by reductive elimination [120]. The effect of the base is more pronounced in achieving the maximum yield. Sodium hydroxide (NaOH) was investigated as an appropriate base for this reaction. In this methodology, the substituted aryl bromide **93** was treated with aryl siloxanes **94** by using magnetically recoverable Pd/Fe_3_O_4_ catalyst and NaOH in H_2_O at 90 °C to obtain the excellent yield (92%) of desired coupling product **95** (Figure 33) [121].

Pd-Fe_3_O_4_ Heterodimeric nanocrystals have been found to be effective and recyclable catalysts to achieve a successful Hiyama cross-coupling under ligand-free conditions. Keeping the effectiveness of catalyst in view, Lee et al. proposed a methodology for the synthesis of diversified biaryl derivatives. The protocol involved the treatment of **96** with phenyl trimethoxysilane **8** under ligand-free conditions. The reaction was proceeded in DMA (dimethylacetamide) using 1 mol% of Pd-Fe_3_O_4_ heterodimeric nanocrystals as a catalyst, KF as a base, and tetra-*n*-butylammonium iodide (TBAI) as an additive at high temperature of 150 °C. Consequently, the desired biphenyl derivative **97** was afforded in 94% yield (Figure 34). Electron-donating and -withdrawing groups (-Me, -OMe, -F, -CF_3_, -NO_2_) bearing aryl halides provided the corresponding biaryl derivatives in an excellent yield range (70–94%) [122].

## 7. *N*-Heterocyclic Carbenes (NHCs) Palladium Complexes

NHCs are nucleophilic in nature as they exhibit σ-electron donating attributes. The application of *N*-heterocyclic carbene (NHC) ligands in transition metal-catalyzed cross-coupling acquired notable interest recently [123,124,125,126]. Peñafiel et al. synthesized [(NHC)_2_PdCl_2_] complex through direct metalation of 0.2% hydroxy-functionalized imidazolium salt and 0.1% palladium acetate to efficiently catalyze the Hiyama cross-coupling of aryl chlorides and bromides with arylsiloxanes under microwave irradiation and fluoride free conditions. The fluoride-free Hiyama reaction of 4-bromoanisole **64** was carried out with phenyl trimethoxysilane **8** using [(NHC)_2_PdCl_2_] complex **98** to catalyze the reaction under microwave irradiation (80 W, 120 °C) for 1 h. NaOH was screened to be a suitable base for the corresponding reaction to afford the targeted derivative **66** in 95% yield (Figure 35). The coupling of aryl and heteroaryl bromides with siloxanes resulted in biaryl products in a 5–88% yield range. The required products were obtained in lower yields with aryl chlorides than with aryl bromides [127]. Another research group of Pastor and coworkers disclosed the efficient synthesis of biaryl derivatives using 0.2 or 0.5% of imidazolium salt in 2013 [128]. 

The synthesis of four linear dinuclear *N*-heterocyclic carbene palladium complexes was reported by Yang and Wang. The structures were characterized by NMR, FT-IR, and elemental analysis. The synthesis of NHC-palladium complexes was achieved by a one-pot reaction of imidazolium salts, PdCl_2_, and bidentate *N*-heterocycles in the presence of potassium carbonate (K_2_CO_3_) with the formula [PdCl_2_(NHC)]_2_(μ-L) (L = DABCO, pyrazine). The catalytic effect of *N*-heterocyclic carbene palladium complexes was investigated in the Hiyama coupling reaction of aryl trialkoxysilanes with aryl chlorides. Among four dinuclear *N*-heterocyclic carbenes, **100** expressed excellent catalytic activity in the corresponding Hiyama coupling reaction. Both electron-donating and electron-withdrawing groups substituted aryl chlorides afforded biphenyl products in (54–92%) yield range. The reaction of **99** with phenyl trimethoxysilane **8** was conducted at 120 °C for 5 h using 0.5 mol% of NHC-Pd **100**, TBAF in toluene to achieve a 92% yield of corresponding product **9** (Figure 36) [129].

The application of homoleptic chelating *N*-heterocyclic carbene palladium complexes immobilized within the pores of SBA-15/IL (NHC-Pd@SBA-15/IL) as heterogeneous catalyst in the Hiyama coupling reaction was reported by Rostamnia and coworkers. The desired biaryl derivatives were achieved in an excellent yield range (82–95%). The catalytic activity of the catalyst was analyzed by treating phenyl trimethoxysilane **8** with phenyl iodide **76** using 0.8 mol% (NHC-Pd@SBA-15/IL) **101**. The reaction proceeded in H_2_O/dioxane (1:2) at 80 °C using Cs_2_CO_3_ as a suitable base and TBAF that increased the catalytic activity of (NHC-Pd@SBA-15/IL). As a result, **92** was obtained in an excellent (95%) yield (Figure 37). Chemoselectivity of (NHC-Pd@SBA-15/IL) for the Hiyama coupling reaction was analyzed by treating *p*-bromobenzaldehyde with phenyl trimethoxysilane under the optimized reaction conditions giving the corresponding biaryl product. Reusability of catalyst for five successful runs was observed [130]. 

Yang synthesized the tetrazole ligand stabilized NHC-Pd complexes by reacting dimeric compounds [Pd(μ-Cl)(Cl)(NHC)]_2_ and tetrazole ligands that catalyzed the Hiyama coupling reaction very well. The catalytic activity of mono and dinuclear Pd(II) complexes were analyzed. Among them, the mononuclear palladium complex [PdCl_2_(SIPr)(1-phenyl-1*H*-tetrazole)] proved to be the most effective one giving a maximum yield (88%) of biaryl product **9**. For this purpose, compound **99** was refluxed with phenyl trimethoxysilane **8** for 8 h using 0.5 mol% of [PdCl_2_(SIPr)(1-phenyl-1*H*-tetrazole)] **102** as catalyst and TBAF in toluene (Figure 38) [131].

A very interesting application of palladium PEPPSI (Pyridine Enhanced Precatalyst Preparation Stabilization and Initiation) complexes with non-bulky NHC ligands as a catalyst precursor in cross-coupling reactions was disclosed by Osińska et al. for the synthesizing the non-symmetric biaryl scaffolds. The application of PEPPSI complexes in Suzuki–Miyaura was investigated, in which their catalytic activity was clearly evidenced [132,133,134,135,136,137], while the use of PEPPSI complexes in the Hiyama coupling reaction is rare [138,139,140]. The PEPPSI complexes were synthesized by reacting a palladium dimer [Pd(bmim-y)X_2_]_2_ and N-ligand. Hiyama coupling worked efficiently in ethylene glycol, and 97% conversion was achieved in the case of coupling of 2-bromotoluene with phenyl trimethoxysilane using Pd(IPr)Cl_2_(Cl-py) **103** (Figure 3) as a catalyst in ethylene glycol solvent. The reaction did not work efficiently in water. Pd(bmim)Br_2_(CN-py) complex **104** catalyzed Hiyama coupling of substituted bromobenzenes and chlorobenzenes with phenyl trimethoxysilane afforded cross-coupled products in (50–93%) yields using NaOH base and ethylene glycol as solvent at 110 °C for 24 h with the exception of 1-chloro-3-nitrobenzene and 4-chloro-2-nitrobenzene giving no results [141]. 

## 8. Nanoparticles as Catalyst

The substantial application of transition metal nanoparticles in catalysis has remained an area of interest for research groups due to their harmless characteristic features, easy preparation methods, and large surface area [142,143]. Srimani et al. synthesized the palladium nanoparticles by the dropwise addition of metal acylate salt (CO)_5_W=C(CH_3_)O(−)NEt_4_(+) solution to the PEG-6000 containing an aqueous solution of K_2_PdCl_4_ to catalyze the Hiyama coupling reactions [144]. It was noticed that the increased amount of PEG-6000 stabilizer decreases the size of nanoparticles. The substituted aryl bromides **105** were coupled with phenyl triethoxysilane **8** using palladium nanoparticles as catalyst **106** and an appropriate base NaOH in the air at 90 °C to achieve the desired biaryls **107** in 98% yield (Figure 39). H_2_O was utilized as an effective solvent in place of THF or DME [145].

A simple, convenient, and rapid one-pot Hiyama coupling of aryl bromides or iodides with arylsilanes under fluoride-free conditions was reported by the research group of Ranu and coworkers. The corresponding reaction was catalyzed by using palladium nanoparticles that were generated in situ from sodium dodecyl sulfate (Na_2_PdCl_4_/SDS) in H_2_O. SDS stabilizes the formation of palladium nanoparticles. The reaction of phenyl bromide **55** with phenyl trimethoxysilane **8** was carried out using palladium nanoparticles that efficiently catalyzed the reaction. Excellent yield (96%) of desired biphenyl product **92** was attained using water as solvent. A series of bases (KOH, Na_2_CO_3_, NaOAc, and NaHCO_3_) was screened, and NaOH was investigated as a suitable base for this reaction (Figure 40). Environmentally friendly solvent, no utilization of ligand, and good yields are the salient features of this synthetic protocol [146].

Diarylmethanes have received appreciable attention from researchers due to their pharmacological activities [147,148,149,150,151]. They are regarded as distinct components of supramolecular structures, including catenanes, macrocycles, and rotaxanes [152,153,154,155]. Focusing on the significance of diarylmethanes, Sarkar and colleagues designed a successful protocol for the synthesis of diarylmethane derivatives using palladium nanoparticles in THF. Palladium nanoparticles were formed by the reaction of 4 mol% K_2_PdCl_4_ with PEG-600 at 70 °C. PEG-600 was used as a reducing and stabilizing agent for the preparation of nanoparticles [156]. Numerous diarylmethane derivatives were formed by Hiyama cross-coupling of benzyl halides with phenyl trialkoxysilanes in a 78–95% yield range. The **108** was coupled with phenyl trimethoxysilane **8** in the presence of 4 mol% K_2_PdCl_4_, PEG-600 using TBAF in tetrahydrofuran at 70 °C under argon environment to afford the desired coupling product **109** in 95% yield (Figure 41). They applied the same methodology to synthesize diarylmethanes **111** in 95% yield using allyl halides **110** and phenyl trimethoxysilane **8** as reacting substrates (Figure 42). The naturally occurring 2,4-bis(4-hydroxybenzyl)phenol **112** was also synthesized by the same research group through multiple steps reactions starting from anisole using the same methodology (Figure 4) [157].

Zhang et al. synthesized a silica-coated SiO_2_@Fe_3_O_4_-Pd catalyst that efficiently catalyzed the Hiyama coupling of **113** with phenyl trimethoxysilane **8**. The catalytic efficacy of the supported catalyst remains even after 10 times of recycling. The reaction proceeded by using SiO_2_@Fe_3_O_4_ supported palladium catalyst (0.5 mol%) using TBAF as a suitable base in THF at 60 °C under an N_2_ atmosphere to obtain desired products **114** in 99% yield (Figure 43). It was noticed that either electron-donating or electron-withdrawing groups containing aryl halides gave a good to excellent yield range (81–99%) [158]. 

Premi and Jain, in 2013, carried out the synthesis of substituted biphenyl derivatives, aromatic heterocycles, and substituted styrene derivatives by phosphane-free Hiyama cross-coupling reaction of aryl and heterocyclic halides with aryl and vinyltrimethoxysilane using palladium nanoparticles in ionic liquids. Palladium nanoparticles were generated by adding the solution of palladium acetate in acetonitrile to 3-(3-cyanopropyl)-1-methyl-1*H*-imidazol-3-ium hexafluorophosphate {[CN-bmim]PF_6_}, an ionic liquid, that stabilizes synthesis of palladium nanoparticles. It was observed that the electron-donating and -withdrawing substituents on arylhalides resulted in an excellent yield range (76–98%). The reaction of iodobenzene **76** with phenyl trimethoxysilane **8** using 4 mol% Pd(OAc)_2_ in **115** and CH_3_CN and 1-butyl-3-methylimidazolium fluoride [bmim]F, an organosilane activator, at 70–120 °C for 8 h to afford the desired biphenyl derivative in 98% yield (Figure 44). The substituted styrene derivatives were also synthesized in (75–98%) yields by the Hiyama cross-coupling of a broad range of aryl iodides with vinyltrimethoxysilane catalyzed by Pd-NPs at 60–70 °C for 15–30 min [159].

In Heck and copper-free Sonogashira reactions, palladium nanocatalysts synthesized by hydrogenation of Pd(dba)_2_ using tris-imidazolium iodide that stabilizes the synthesis of nanocatalysts are substantially used. The catalytic efficacy of the corresponding nanocatalyst regarding Hiyama coupling reaction under fluoride-free conditions was reported by Planellas et al. in 2014. The authors synthesized the nanoparticles by hydrogenation of Pd(dba)_2_ [160] using tris-imidazolium iodide [161] and adopted an impressive approach for the synthesis of substituted styrene derivatives through Hiyama coupling reaction of substituted aryl iodides **116** with triethoxy(vinyl)silane **10** using nanocatalyst **117** (0.25 mol% Pd catalyst loading). NaOH was used as a suitable base in a 1:1 mixture of MeOH/H_2_O. The reaction worked effectively at 100 °C to attain **118** in 97% yields (Figure 45). The synthesis of unsymmetrically-substituted stilbene **120** was achieved in 82% yield through a one-pot Hiyama–Heck reaction of 1-iodo-4-nitrobenzene **7**, triethoxy(vinyl)silane **10**, and 1-iodo-4-methoxybenzene **119** catalyzed by **117** in the presence of NaOH in MeOH/H_2_O (1:1) (Figure 46) [162].

The catalysis of a facile, cost-effective, and ligand-free Hiyama cross-coupling by functionalized SBA-15 palladium nanoparticles was reported by Huang et al. The Pd catalysts (Pd@M-SBA-15 and Pd@P-SBA-15) gave (13–92%) yield range for the Hiyama coupling reaction of a variety of aryltriethoxysilanes with haloaryls. The catalytic efficacy of Pd@M-SBA-15 and Pd@P-SBA-15 is associated with the presence of TMS or TPS groups on mesopores. The reaction of **11** with phenyltriethoxysilane **16** was carried out by utilizing Pd@M-SBA-15 (0.5 mol%) catalyst, acetic acid, and TBAF**^.^**3H_2_O in toluene at 100 °C in the air for 24 h resultantly afforded the corresponding biphenyl derivative **38** in 92% yield (Figure 47). Between 38–89% yields were obtained in the case of coupling of various aryl bromides with aryltriethoxysilanes [163].

*Euphorbia thymifolia* L. belongs to the family of *Euphorbiaceae* and is an apparently small and branched medicinal herb. It exhibits numerous pharmaceutical applications against dysentery, venereal diseases, and diarrhea. Focusing on the importance of Pd NPs, Nasrollahzadeh and colleagues developed a sustainable protocol for the formation of Pd NPs using an aqueous leaf extract of *Euphorbia thymifolia* L. due to its reducing and stabilizing abilities and observed its effectiveness in ligand-free Stille and Hiyama cross-coupling reactions in a green solvent. The Pd-NPs were distinguished by TEM, powder XRD, and UV-visible techniques. The catalytic efficacy of Pd NPs was evaluated for the Hiyama coupling reaction of **121** with phenyl trimethoxysilane **8** to afford the targeted biaryl products **122** in 96% yields (Figure 48). The optimized reaction conditions (1 mol% of Pd NPs, NaOH in H_2_O as solvent at 90 °C, under air) were utilized for the corresponding reaction. Green solvent, high yields, substrate scope, cost-effectiveness, mild reaction conditions, and ease of handling are certain salient features of the corresponding methodology [164].

A homogeneous catalytic system is frequently used in metal-catalyzed C-C coupling reactions. However, due to some drawbacks of a homogeneous catalytic system, including availability, stability, difficult separation, and recycling, diverted the attention of researchers toward the synthesis of heterogeneous catalytic systems. Nanoscale palladium supported on zinc oxide was formed by a coprecipitation method and characterized by XRD, XPS, SEM, TEM, and thermogravimetric analysis. The application of palladium supported on zinc oxide nanoparticles as a novel heterogeneous catalyst for the formation of unsymmetrical biaryl derivatives by Suzuki–Miyaura and Hiyama coupling reaction was reported by Hosseini-Sarvari and colleagues in 2015. Suzuki–Miyaura coupling of aryl halides and aryl boronic acid catalyzed by Pd/ZnO nanoparticles afforded biaryl derivatives in an 84–97% yield range. Hiyama coupling reaction of iodobenzene **76** and phenyl trimethoxysilane **8** resulted in 96% yield of biaryl product **92**. The reaction was catalyzed by Pd/ZnO nanoparticles using K_2_CO_3_ as the base in ethylene glycol under an air atmosphere by maintaining a temperature of 100 °C (Figure 49). Aryl iodides having electron-rich and poor groups gave the desired products in good to excellent yields. Electron deficient aryl bromides and chlorides resulted in biaryl products in short times (40–70 min) [165].

Ohtaka and coworkers developed a pathway for the synthesis of substituted biphenyl derivatives by fluoride-free Hiyama coupling reaction of aryl bromides with aryl trimethoxysilanes and observed the catalytic effectiveness of linear polystyrene-stabilized PdO nanoparticles [PS-PdONPs] and polystyrene-stabilized Pd nanoparticles [PS-PdNPs] under green conditions. For example, the reaction of 4-methylphenylbromide **123** was performed with phenyl trimethoxysilane **8** using PS-PdONPs (1.5 mol%) catalyst, TBAC in aqueous NaOH solution at 80 °C under aerobic conditions for 3 h, which afforded 4-methylbiphenyl **79** in 88% yield. The catalytic effect of PS-PdNPs on the Hiyama coupling reaction was observed, and it was noticed that the desired coupled product was not attained, but 4,4′-dimethylbiphenyl **124**, the Ullmann coupling product, was afforded in 99% yield (Figure 50). PS-PdONPs showed higher catalytic efficacy in contrast to PS-PdNPs. The catalyst was retrieved and successively subjected to four runs of cross-coupling reaction [166].

The Heck and Hiyama coupling reactions clasp a pronounced emplacement due to their implementation in the synthesis of complex organic structures [167,168,169,170]. Focusing on the green methodology, Gaikwad et al. utilized Triton X-100, a non-ionic surfactant, as a stabilizer [171,172] for the generation of palladium nanoparticles and executed an appreciable protocol for the synthesis of symmetrical stilbenes via one-pot sequential Hiyama–Heck coupling reactions. For this purpose, the research group adopted arenediazonium salt and triethoxy(vinyl)silane as substrates catalyzed by Triton X-100 stabilized palladium nanoparticles. The arenediazonium tetrafluoroborate **125** was allowed to couple with triethoxy(vinyl)silane **10** to afford the symmetrical *trans*-stilbene derivative **126** in excellent yield (95%) (Figure 51). The potency of other surfactants, including cetyltrimethylammonium bromide (CTAB) and sodium dodecyl sulfate (SDS), was examined and found Triton X-100 an appropriate non-ionic surfactant for the respective reaction. The reaction was conducted at room temperature by using Triton X-100 (5 mol%) with Pd(OAc)_2_ (2 mol%) in H_2_O [173].

Black pepper (*piper nigrum*) belongs to the *Piperaceae* family. Black pepper extract contains a variety of phytochemicals such as piperine, phenols, ethyl piperonyl cyanoacetate, *N*-isobutyl-tetradeca-2,4-dienamide that help to achieve the reduction of Pd(II) to Pd(0). It is used as an important component in traditional medicines [174]. The formation of green nanocatalyst (Pd NPs) using aqueous ethanolic extract of black pepper was carried out by Kandathil et al. in 2018, and its catalytic activity in the cyanation and Hiyama cross-coupling was observed. The cyanation of aryl halides was carried out in the presence of cyanating reagent K_4_Fe(CN)_6_. Either electron-donating or electron-withdrawing groups substituted on aryl halides gave moderate to excellent yields. The ligand-free Hiyama cross-coupling reaction of various aryl halides with aryl trimethoxysilane was carried out under fluoride-free conditions to afford the biphenyl derivatives in an 87–98% yield range. The reaction of iodobenzene **76** and phenyl trimethoxysilane **8** proceeded at 100 °C in the presence of NaOH, ethylene glycol, and Pd nanoparticles with 0.2 mol% Pd loading, which catalyzed the reaction efficiently, to obtain the cross-coupling product **92** in 98% yield (Figure 52) [175].

## 9. Pd(PPh_3_)_4_ as Catalyst

Cyclopropane motifs have gained huge interest due to their unique features and immense applications in chemical transformations [176,177]. These motifs exist in numerous biologically active natural products and synthetic drugs [178,179,180]. The contribution of the silanol group in the cyclopropanation and Hiyama–Denmark cross-coupling reaction was described by Beaulieu et al. Di-*tert*-butoxy(cyclopropyl)silanol serves as a substrate for Hiyama–Denmark cross-coupling reaction was synthesized by Simmons–Smith via cyclopropanation reaction of di-*tert*-butoxy(alkenyl)silanol. (Cyclopropyl)silanol **127** was treated with BF_3_**^.^**OEt_2_ followed by the reaction with **128**, 5 mol% Pd(PPh_3_)_4_, and TBAF using THF solvent. The reaction worked well at 100 °C to afford the desired cross-coupling product **129** in 92% yield (Figure 53) [181].

In 2015, the synthesis of unsymmetrical biaryl derivatives through the Hiyama cross-coupling protocol in the presence of Cu(I) and H_2_O was reported by Delpiccolo and coworkers. Cu(I) salts have been proven to improve the efficacy of palladium-catalyzed cross-coupling reactions [182]. A series of biaryl derivatives was achieved in a 62–98% yield range. Maximum yield (98%) of **131** was observed in the case of coupling of **130** with **31** using Pd(PPh_3_)_4_ (0.025 equiv.), TBAF as fluoride source and CuI to improve the catalytic reaction in THF (5% H_2_O). The reaction proceeded well at 80 °C for 18 h (Figure 54) [183]. 

## 10. Copper Catalyzed Hiyama Cross-Coupling Reactions

Hiyama cross-coupling remained a valuable tool for the construction of carbon–carbon bonds in diversified natural products, pharmaceutical compounds [184,185,186,187,188,189], and complex structures. Hiyama coupling reaction is predominantly conducted with palladium catalysts. In 2013, Gurung et al. reported the first Cu^I^-catalyzed Hiyama cross-coupling reaction of aryl- and heteroaryl halides with aryl- and heteroaryltriethoxysilane in the presence or absence of PN-1 bidentate ligand. The electron-rich and electron-deficient groups substituted on aryl halides and aryltriethoxysilanes gave moderate to excellent yields. The coupling reaction of heteroaryl triethoxysilanes with aryl iodides in the presence of PN-1 ligand afforded the desired biaryl derivatives in a 40–74% yield range. An excellent result (94%) of **134** was obtained by the addition of **132** to **133** using 10 mol% CuI and CsF in the absence of PN-1 ligand **135**. The reaction was conducted in DMF by maintaining the temperature at 120 °C for 24 h (Figure 55). CsF stabilizes the formation of monomeric [CuAr] intermediate and acts as a fluoride source for the corresponding reaction [190].

Allylbenzenes have acquired an eminent interest in organic synthesis due to their pharmacological activities. The research group of Cornelissen carried out the synthesis to afford allylbenzene derivatives via copper-catalyzed Hiyama cross-coupling of vinylalkoxysilanes in the presence of an activating agent, TBAT (tetrabutylammonium difluorotriphenylsilicate) without using any ligand. Various allylbenzene derivatives were achieved in an 83–99% yield range by reaction of vinyl silane with substituted benzyl halides. The substrate **136** was allowed to couple with **137** to afford the desired product **138** in maximum (99%) yield. The optimization reaction conditions were Cu[MeCN]_4_PF_6_ (10%), TBAT, MeCN solvent, 40 °C, 16 h (Figure 56). The benzylation of various vinylsilanes gave coupling products in a 51–92% yield range. Moreover, *Z*-alkenes were synthesized using a copper-catalyzed Hiyama cross-coupling reaction. The substrate *β*-(*Z*)-vinylsilane **139** was coupled with **140** to acquire the desired *Z*-alkene **141** in 92% yield under optimized reaction conditions (Figure 57) [191].

## 11. Nickel-Catalyzed Hiyama Cross-Coupling Reaction

Bimetallic nanoparticles have attained substantial attention as an efficient catalytic system for Hiyama cross-coupling reactions. Rothenberg and coworkers synthesized the core–shell Ni-Pd nanoclusters by combining electrochemical and wet chemical methods that efficiently catalyzed Hiyama coupling reactions in contrast to bimetallic alloy clusters and monometallic clusters. Aryl iodides with electron-donating and electron-withdrawing substituents gave substituted biaryl derivatives in a 6->99% yield range. A competent reaction of haloaryls **142**, phenyl trimethoxysilane **8**, and **143** afforded the corresponding biaryl product **144** in good (>99%) yield along with homocoupling side-product (<2%). The reaction worked efficiently at 65 °C in the presence of core–shell Ni-Pd cluster (1 mol%) catalyst and tetrabutylammonium fluoride (TBAF) under an N_2_ atmosphere. THF was utilized as the only solvent for the corresponding reaction (Figure 58) [192].

The nickel/bathophenanthroline catalyzed Hiyama coupling reaction of unactivated secondary alkyl halides has been recently reported, but it was not found to be an efficient catalyst for activated secondary alkyl halides. The activity of amino alcohol ligand was also observed in the case of the Hiyama coupling reaction. Later, in 2007, the catalytic effect of nickel/norephedrine in Hiyama coupling reactions of several unactivated alkyl halides was reported by Strotman et al. The cyclohexyl iodide **146** was treated with phenyl trifluorosilane **147** to achieve the desired cross-coupling product **148** in 94% yield. The reaction worked efficiently at 60 °C in the presence of 10% NiCl_2_**.**glyme with 15% norephedrine as a catalyst, using 12% lithium hexamethyldisilazide (LiHMDS) as the base, H_2_O and CsF in DMA (*N*,*N*-dimethylacetamide) (Figure 59). In the case of nickel-catalyzed Hiyama coupling reactions of activated secondary alkyl halides, excellent results were obtained in a 60–92% yield range under the same reaction conditions mentioned above [193].

Dai et al. reported a convenient protocol for Ni-catalyzed asymmetric Hiyama cross-coupling reactions of racemic α-bromo esters. On varying reaction conditions, several α-aryl esters were obtained in a low to excellent yield range (<2–92%) with enantioselectivities (13–99%), respectively. The phenylation of *α*-bromo ester **149** was carried out with phenyl trimethoxysilane **8** at room temperature using 10% NiCl_2_**^.^**glyme, 12% **150**, and TBAT in dioxane yielded the corresponding coupling product **151** in 80% yield with 99% *ee* enantioselectivity (Figure 60). The addition of substituted aryl silanes to α-bromo esters resulted in 64–76% yields of respective cross-coupling products and 87–94% *ee* enantioselective values, while α-bromo esters underwent alkenylation through catalytic asymmetric Hiyama coupling reactions afforded the coupling products in 66–72% yield range with 91–93% *ee* [194].

The environmentally benign Hiyama cross-coupling of tetrafluoroethylene (TFE) and perfluoroarenes via C-F bond activation was developed by Ogoshi and coworkers in 2014. An excellent yield (90%) of **154** was obtained when **152** was treated with **153** using 5 mol% [Ni_2_(*^i^*Pr_2_lm)_4_(cod)] catalyst in tetrahydrofuran at 100 °C for 10 h (Figure 61). The palladium-catalyzed base-free Hiyama coupling of tetrafluoroethylene with substituted aryl trimethoxysilane yielding *α*,*β*,*β*-trifluorostyrene derivatives in 40–94% yield range. The reaction was carried out using 2.5 mol% Pd_2_(dba)_3_(C_6_H_6_) as the catalyst, 5 mol% PCyp_3,_ and 10 mol% FSi(OEt)_3_ in THF at 100 °C [195].

Wang and coworkers carried out the fluoroalkylation of arylsilanes via nickel-catalyzed Hiyama coupling reactions. The arylsilanes bearing electron-rich substituents underwent monofluoroalkylation smoothly to afford excellent yields (74–94%). The arylsilane **155** was coupled with **156** using a catalytic amount of Ni(dme)Cl_2_ (10 mol%), **157** (12 mol%) CsF as an activator to achieve the desired monofluoroalkylated product **158** in 94% yields. 1,4-Dioxane was the only solvent used for this reaction, and the temperature was maintained at 80 °C (Figure 62). Ezetimibe is a cholesterol absorption inhibiting drug [196]. The monofluoroalkylation of ezetimibe-derived arylsilane proceeded under optimized reaction conditions to afford the monofluoroalkylated ezetimibe **159** in 91% yield (Figure 5) [197].

The palladium-catalyzed decarboxylative coupling reactions have been reported by many research groups in the past. The nickel-catalyzed decarboxylative Hiyama coupling reactions of alkynyl carboxylic acids with organosilanes were introduced for the first time by Raja et al. The reaction of **160** was carried out with phenyl triethoxysilane **16** using a catalytic amount of Ni(acac)_2_ (10 mol%), affording the corresponding decarboxylative product **161** in 90% yield along with undesired homocoupling product **162** in low yield. The application of 1,10-phenanthroline as a ligand provided good results by using CsF as an activator of organosilanes and CuF_2_ as an oxidant. The reaction worked well in DMSO at 120 °C for 12 h (Figure 63). Various aryl alkynyl carboxylic acids were allowed to couple with substituted phenyl triethoxysilanes, resulting in decarboxylative products in a 74–90% yield range. A 90% yield of the desired decarboxylative product was also obtained in the case of *p*-chloro phenyl triethoxysilane. The symmetrical diarylacetylenes were synthesized in a 21–45% yield range under optimized reaction conditions. It was noticed that variation in reaction conditions (in the presence of TEMPO and the absence of CuF_2_, CsF, and Ni ligands) resulted in relatively low yields of corresponding products [198].

## 12. Applications of Hiyama Coupling Reactions for the Synthesis of Biologically Active Scaffolds

Retinoids, chemically related to vitamin A, have been the subject of tremendous endorsement due to their activities in biological processes, including cell growth, vision, embryonic development, immune response, and reproduction [199]. The transition metal-catalyzed Suzuki and Stille cross-coupling reactions have been extensively utilized to obtain retinoids. However, drawbacks to the above reactions, including low stability of organoboranes, toxicity, and high molecular weight of organostannanes, diverted the attention of researchers towards the highly efficient Hiyama coupling reaction. The total synthesis of retinoids was first outlined by Montenegro et al., employing the Hiyama coupling reaction as a key step. To the solution of organosilane reagents **164** and **167** in THF, TBAF was added, followed by the addition of trienyl iodide **163** and Pd_2_(dba)_3_**^.^**CHCl_3_ to afford retinyl ethers **165** and **168**, which underwent deprotection using TMSCl, H_2_O, and methanol to attain *trans*-retinol **166** and 11-*cis*-retinol **169** in 74% and 83% yields, respectively. The oxidation of 11-*cis*-retinol in the presence of BaMnO_4_ in dichloromethane solvent afforded the synthesis of 11-*cis*-retinal **170** in 90% yield (Figure 64) [200].

Heliannuol A, isolated from sunflower *Helianthus annuus* [201,202,203,204,205,206], is regarded as the first member of the family of allelopathic [207,208] sesquiterpenoids holding benzoxocane moiety. Heliannuols have attained eminent attention from synthetic chemists due to their possessing unusual structures and biological activity. Vyvyan and coworkers described the synthesis of benzoxocane using trisubstituted *Z*-styrene derivatives synthesized by Hiyama coupling of oxasilacycloalkenes with aryl iodides. The substituted *Z*-styrene derivative, obtained by Hiyama coupling of **171** and **172** using Pd_2_(dba)_3_ (2–3 mol%) catalyst and tetrabutylammonium fluoride (TBAF) in THF solvent, underwent intramolecular Buchwald–Hartwig etherification using 10 mol% Pd_2_(dba)_3_ catalyst and 10 mol% Q-Phos ligands at 80 °C for 24 h. NaO*t*-Bu was selected as a suitable base in toluene for corresponding etherification reaction to afford the maximum (10%) yield of benzoxocane **174** along with **175** as the major product (Figure 65). The hydrogenation of benzoxocane **174** using Pd/C in ethanol gave **176** in 44% yield [209].

The synthesis of biologically relevant benzofuran scaffolds by a convenient route is of huge interest because benzofuran compounds account for important bioactive molecules, including amiodarone [210], BNC105 [211], cytotoxic flavonoids [212], and natural products such as egonol [213,214], daphnodorin A and B [215] and moracin O and P [216,217]. The application of palladium(II) acyclic diaminocarbene (ADC) complexes in one-pot tandem Hiyama alkynylation/cyclization for synthesizing biologically relevant benzofuran derivatives was disclosed by Singh et al. The palladium ADC complexes were synthesized via nucleophilic addition of secondary amines such as morpholine, pyrrolidine, and piperidine to metal precursor *cis*-{(2,4,6(CH_3_)_3_C_6_H_2_)NC}_2_PdCl_2_ at room temperature that efficiently catalyzed synthesis of benzofuran derivatives. Iodophenol and triethoxysilylalkynes underwent Hiyama alkynylation followed by cyclization catalyzed by Pd ADC complex to afford benzofuran derivatives in (14–57%) yield. Excellent result (57%) of benzofuran derivative **180** was obtained by Hiyama alkynylation/cyclization of **177** and **178** using 2 mol% palladium ADC complex **179** as catalyst and NaOH as the base. The reaction proceeded in the 4:2 mixture of 1,4-dioxane/H_2_O by maintaining the temperature at 80 °C for 4 h. The Hiyama cross-coupling of iodobenzene with triethoxysilylalkynes catalyzed by palladium ADC complex **179** gave alkyne derivatives in (35–76%) yield range using optimized reaction conditions (Figure 66) [218].

Triflate derivatives have remained an area of interest for researchers due to their portfolio in catalytic transformation, i.e., one-pot tandem Heck alkynylation/cyclization reactions [219], and in biomedical applications as potent anticancer agents [220]. The utility of triflate derivatives of palladium acyclic diaminocarbene (ADC) complexes as effective precatalysts for Hiyama alkynylation/cyclization reaction in the synthesis of benzofuran derivative was reported by Ghosh and coworkers. The complexes cis-[(R^1^NH)(R^2^)methylidene]Pd(OCOCF_3_)_2_(CNR^1^) [R^1^ = 2,4,6-(CH_3_)_3_C_6_H_2_: R^2^ = NC_4_H_8_ (**181**); NC_5_H_10_ (**182**)] were synthesized by the treatment of chloro derivatives *cis*-[(R^1^ NH)(R^2^)methylidene]PdCl_2_(CNR^1^)[R^1^ = 2,4,6- (CH_3_)_3_C_6_H_2_: R^2^ = NC_5_H_10_; NC_4_H_8_] with AgOCOCF_3_ in excellent yields (84–94%) under ambient conditions. One-pot tandem Hiyama alkynylation/cyclization reaction of iodophenol with a range of triethoxysilylalkynes catalyzed by palladium acyclic diaminocarbene triflate complexes gave low to moderate yields of corresponding benzofuran derivatives. It was observed that the application of precatalyst **182** exhibited a comparatively higher yield range (15–52%) than precatalyst **181** (10–49%). The essential parameters for the smooth functioning of the corresponding reaction comprised 2 mol% precatalysts **181** and **182**, NaOH in a 4:2 mixture of 1,4-dioxane:H_2_O at 80 °C for 4 h. Maximum yields of 49% and 52% were obtained in case of coupling of **177** with **178** catalyzed by palladium acyclic diaminocarbene (ADC) complexes **181** and **182** precatalysts, respectively (Figure 6) [221].

## 13. Miscellaneous

The Hiyama coupling of unactivated alkyl bromides and iodides in mild conditions was achieved by Lee et al. A series of cross-coupled products were achieved in a 65–81% yield range by Hiyama coupling of functionalized alkyl bromides with phenyl trimethoxysilanes using PdBr_2_ catalyst, P(*t*-Bu)_2_Me and Bu_4_NF. The reaction was carried out in THF at room temperature, while in the case of [HP(t-Bu)_2_Me]BF_4_, 42–88% yields of cross-coupled products were achieved. The **183** was allowed to couple with **184** using 4% PdBr_2_/P(*t*-Bu)_2_Me and Bu_4_NF in the presence of THF solvent at room temperature to afford the desired coupling product **185** in 84% yield (Figure 67). The same Hiyama coupling reaction also proved to be effective for functionalized alkyl iodides [222].

Alacid and Nájera reported the NaOH promoted Hiyama coupling reactions of aryl halides with vinyltrialkoxysilanes using fluoride-free conditions. The Hiyama coupling of vinyltrimethoxysilane with aryl bromides and chlorides afforded styrene derivatives in 47–99% yields. The reaction of 4-bromo acetophenone **11** was carried out with vinyltrimethoxysilane **50** using 4-hydroxyacetophenone oxime-derived palladacycle (0.1 mol% Pd), 2.5 equivalent NaOH promotor, and 1 equivalent of tetra-*n*-butylammonium bromide (TBAB) additive under microwave irradiation for 10 min. In the case of styryltriethoxysilane, stereospecific coupling with aryl or vinyl bromides provided the corresponding stilbenes or dienes, respectively. Moreover, the undesirable polymerization of products is prevented by using mild reaction conditions in this protocol (Figure 68) [223].

Bhaumik and coworkers synthesized the palladium containing periodic mesoporous organosilica (PMO) and analyzed their catalytic efficacy in Hiyama coupling, Sonogashira coupling, and cyanation reaction. Despite Sonogashira coupling and cyanation reaction, the focus was being placed on environment-friendly Hiyama coupling to achieve unsymmetrical biphenyls. The substituted benzonitriles and disubstituted alkynes were achieved in 68–95%, and 72–90% yield ranges, respectively. Hiyama coupling reaction proceeded under green conditions, using a Pd-containing (PMO) catalytic system giving 60–95% yield of the respective coupled products. The best results were obtained in the case of coupling reaction of *p*-iodonitrobenzene **7** with trimethoxy(vinyl)silane **50** to attain a 95% yield of the targeted product **187**. The essential parameters utilized in the reaction comprised of Pd-LHMS-3 catalytic system and NaOH at 100 °C in water (Figure 69). The excellent yields of products, reusability, and easy work-up made Pd-grafted PMO an effective catalyst for synthesizing benzonitrile derivatives, disubstituted alkynes, and unsymmetrical biphenyls [224].

The catalytic activity of thermally stable and oxygen insensitive dimeric ortho-palladated complex [Pd{C_6_H_4_(CH_2_N(CH_2_Ph)_2_)}(µ-Br)]_2_ homogeneous catalysts [225] in Hiyama coupling was investigated by Hajipour and colleagues. The substituted biphenyls were obtained in a 68–97% yield range along with homocoupling products. Maximum yield (97%) of **92** was observed in the case of coupling of aryl halides (bromides and iodides) **188** with phenyl triethoxysilane **16** using efficient [Pd{C_6_H_4_(CH_2_N(CH_2_Ph)_2_)}(µ-Br)]_2_ catalysts under microwave irradiation at 90 °C and 500W. Among different solvents (THF, DMF, Dioxane, EtOH, MeOH, *p*-Xylene), DMF was screened to be a microwave-active polar solvent and TBAF**^.^**3H_2_O as an efficient additive to improve the yield within short times (Figure 70) [226].

Arylsulfonyl chlorides have been used for many years in manufacturing pesticides, dyes, drugs, polymers [227,228], etc., due to their affordability and ease of availability. The application of arylsulfonyl chlorides in palladium-catalyzed desulfitative Hiyama coupling to afford biphenyl derivatives under mild reaction conditions was outlined by Zhang et al. The reaction of phenylsulfonyl chloride **191** with 4-methoxyphenyltrimethoxysilane **192** underwent palladium-catalyzed desulfitative Hiyama coupling by using Pd_2_(dba)_3_ (3 mol%) catalyst and TBAF**^.^**3H_2_O as an additive to obtain the desired biaryl derivative **66** in 92% yield (Figure 71). The 1:4 mixture of DMF/CH_3_CN was utilized as a suitable solvent for the corresponding reaction by observing the results of DMSO, DMF/1,4-dioxane, NMP, THF, and toluene. The reaction worked well at 100 °C for 3 h under an N_2_ atmosphere [229].

A novel method for the formation of unsymmetrical biaryls using arylsulfonyl hydrazides by Hiyama cross-coupling was developed by Miao et al. The arylsulfonyl hydrazides containing electron-rich and -poor groups smoothly underwent cross-coupling with phenyl trimethoxysilane providing a series of biaryl derivatives in a 72–95% yield range. The coupling of **193** with phenyl trimethoxysilane **8** catalyzed by 5 mol% Pd(TFA)_2_ gave the desired biaryl product **66** in 95% yield under desulfitative and denitrificative reaction. The reaction worked efficiently in DMI solvent using a TBAT (tetrabutylammoniumdifluorotriphenylsilicate) activator at 60 °C for 12 h under an O_2_ atmosphere (Figure 72). Between 74–94% of yields of desired cross-coupled products were obtained by the addition of phenylsulfonyl hydrazide with a wide range of aryl trimethoxysilanes under optimized conditions [230].

In 2016, the catalysis of Suzuki–Miyaura and Hiyama–Denmark coupling of aryl sulfamates using (1-^t^Bu-indenyl)Pd(L)(Cl) precatalysts was reported by Hazari and coworkers. Hiyama–Denmark coupling of substituted aryl silanolates and aryl chlorides afforded the desired coupled products in maximum yields (85–97%). A total of 2.5 mol% Pd-P^t^Bu_3_ precatalysts, toluene, 70 °C, and 4 h were the standard reaction conditions (Figure 73). The same research group described the first Hiyama–Denmark coupling of aryl sulfamates using Pd-RuPhos precatalyst. Several additives (NaOMe, NaO*^t^*Bu, and RuPhos) were screened to improve the yields of Hiyama–Denmark reactions. An excellent result (91%) was obtained in the case of coupling of **197** with **198** in the presence of 5 mol% Pd-RuPhos precatalyst and 5 mol% RuPhos as an additive in toluene at 110 °C for 8 h (Figure 74) [231].

The synthesis of valuable biarylboronates via photochemical gold-catalyzed Hiyama redox-neutral arylation of mechanistically preferential B,Si-bimetallic coupling reagents with substituted diazonium salts was reported by Hashmi and coworkers. The scope of Hiyama coupling with respect to boronic acid derivatives was investigated. It was observed that boronic acid derivatives (BMIDA, BPin, and Bnep) with substituted aryl silanes afforded desired cross-coupled products in reasonable (49–86%), (40–65%) and (42–60%) yield ranges, respectively. Excellent output (91%) of desired biofunctionalized biarylboronate derivative **202** was attained in case of coupling of B,Si-bimetallic reagent **201**, and 4-CF_3_ substituted aryldiazonium salt **200** using a 10 mol% Ph_3_PAuNTf_2_ catalyst in acetonitrile solvent under irradiation of blue LEDs at room temperature (Figure 75) [232].

A very interesting green procedure of Hiyama cross-coupling using a highly stable NCN-pincer palladium complex as catalyst was disclosed by Marset et al. The catalytic efficacy of the NCN-pincer palladium complex was assessed during the coupling of aryl iodides and bromides and phenyl trimethoxysilane. The aryl bromides and iodides with electron-donating or electron-withdrawing groups gave moderate to excellent yields. The NCN-pincer palladium complex catalyzed coupling of aryl iodides and bromides with phenyl trimethoxysilane in glycerol gave expected biaryl products in 40–99% yields while a 1:2 mixture of choline chloride:glycerol (ChCl:glycerol) facilitated the synthesis of targeted biaryl products in 1–85% yields. Maximum yield (99%) of biphenyl product **92** was obtained in the case of coupling phenyl iodide **76** with phenyl trimethoxysilane **8** using 1 mol% NCN-pincer palladium-catalyzed catalyst **203** in 0.5 M glycerol at 100 °C for 24 h. K_2_CO_3_ was screened as a suitable base for the corresponding reaction (Figure 76). Isomerization was observed in the case of reaction with allyl trimethoxysilane. A mixture of isomers was formed, affording a highly stable internal *trans* double bond containing a major product in 83% yield. The recyclability of catalysts and the usage of biorenewable solvents are the salient features of this methodology [233].

Sobhani and coworkers synthesized a hydrophilic heterogeneous cobalt catalyst supported on chitosan [mTEG-CS-Co-Schiff-base]. The catalyst was identified by XRD, TEM, FT-IR, TGA, ICP, FE-SEM, and XPS analyses, and its implication as a heterogeneous catalyst in Hiyama, Heck, Suzuki, and Hirao cross-coupling was investigated. The application of the mTEG-CS-Co-Schiff base in fluoride-free Hiyama coupling reaction under green reaction conditions was reported for the first time. A series of biaryl derivatives having electron releasing and electron-withdrawing groups on aryl halides were synthesized in a good to excellent yield range (80–98%). The coupling reaction of iodobenzene **76** with triethoxyphenylsilane **16** worked well under green conditions using 0.5 mol% mTEG-CS-Co-Schiff-base (catalyst) **204** and NaOH as a base. Consequently, an excellent (98%) yield of desired biaryl product **92** was obtained (Figure 77). The utilization of water as a green solvent, recovery, and scalability of catalyst for at least six successful runs without losing catalytic activity and inexpensive abundant cobalt catalyst make this reaction an environmentally and economically friendly protocol [234].

The comparison of catalysts reported in the section above indicates that among the palladium-catalyzed reaction, several Pd based catalysts afforded the products in quantitative yields (Table 1, entries 1, 2, 3, 4, 5, 7, 11, 12, 22, 23, 24, 25, 40 and 41) [53,54,55,56,64,81,84,95,96,97,104,141]. Among these Pd catalyzed methodologies, [PdCl_2_P(OCH_2_CMe_2_NH)OCH_2_CMe_2_N_2_] at catalyst loading of 1.31 × 10^−5^ mol was found to afford a range of products in excellent yields (87–99%) at mild reaction conditions (Table 1, entry 25) [104]. Further, the catalyst Pd(dppf)Cl_2_ (0.05 equiv), Pd(OAc)_2_/**17**, Pd(OAc)_2_/**26** afforded products in promising yields of 72–99%, using water as green solvent (Table 1, entry 23) [96] and 70–98%, and 44–99% under neat and mild reaction conditions, respectively (Table 1, entries 5 and 8) [56,64]. The catalyst Pd(allyl)Cl]_2_/**47** at 1.25% catalyst loading efficiently yielded the products in an excellent to quantitative yield range of 90–95% at two different temperatures, 110 °C and 115 °C, using THF solvent (Table 1, entry 15) [88]. However, some catalysts, such as Pd(Oac)_2_ catalyst and Pd(Oac)_2_/**26**, showed considerable variability in the yield range of 20–100% and 36–97%, respectively (Table 1, entries 1 and 7) [53,64]. Affording the products in poor/moderate to quantitative yields suggested the limited applicability of the catalyst and narrow substrate scope. Among the nanoparticle-based catalysts, SiO_2_@Fe_3_O_4_-Pd at a low catalyst loading of 0.5 mol% of Pd and mild reaction conditions afforded the best yield range (81–99%) (Table 1, entry 56) [158]. While, the Pd NPs **106** (1 mol%) and Pd NPs **117** (0.25 mol% Pd) using green solvent water, readily available base NaOH afforded promising yields of 88–98% (at 90 °C) (Table 1, entry 53) [145] and 59–97% (100 °C) (Table 1, entry 58) [162], respectively. While, Pd NPs (4 mol%) derived from Pd(OAc)_2_ using also afforded promising yields of 76–98%, respectively. Among Copper catalysts, Cu[MeCN]_4_PF_6_ at (10% catalyst loading) yielded products in 83–99% (Table 1, entry 67) [191]. Furthermore, Ni(dme)Cl_2_/**157** at 10 mol% loading and CsF obtained products in excellent yield range 74–94% (Table 1, entry 71) [197]. Overall, the palladium catalyst showed the best results, and due to the yield range, green reaction conditions, and substrate scope, they observed wide applicability for the synthesis of a range of scaffolds.

## 14. Conclusions

In conclusion, we have reviewed a large number of strategic approaches published about Hiyama cross-coupling reactions during the last 15 years. Moreover, this review article highlights all major advances in Hiyama coupling reaction regarding different catalytic systems *N*-heterocyclic carbene (NHC)-Pd complexes, palladium-supported nickel catalysts, copper iodide, nanoparticles as catalysts, etc.), the effect of different substituents on product yields, enantioselectivity, and applications in the synthesis of valuable scaffolds. Transition metal-catalyzed coupling reactions have been emerging as efficient and reliable methodologies to realize the synthesis of a range of products. Transition metal-catalyzed Hiyama reaction is also a coupling reaction yielding scaffolds with potential applications in pharmaceutical and chemical industries. Although a number of methodologies highlighting different transition metal-based catalysts at different catalyst loadings have been reported in recent years, the area needs to be explored more and unbox certain limitations. Several nanocatalysts have been developed that realize the synthesis more efficiently due to higher surface-to-volume ratio and contact ratio with substrates; however, more efficient catalysts with environmentally benign methodologies, mild reaction conditions, and broad substrate scope are desired. Along with the monometallic catalytic systems, the focus could be placed on the development of bi-metallic and tri-metallic catalysts. The nanocatalysts are needed to be explored to develop cheap, reliable, and most efficient catalysts. The substrate scope of the reaction should be extended for the use of aryl chloride along with aryl bromide/iodide as coupling partners. We believe that this review will be a significant contribution to increasing the research interest in the respective field.

## Data Availability

Not Applicable.

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
