# Peer review of "Transition Metal Catalyzed Hiyama Cross-Coupling: Recent Methodology Developments and Synthetic Applications"

_molecules, 2022, doi:10.3390/molecules27175654_

Round 1
Reviewer 1 Report
Present work is devoted to review of . In general, it is useful material for wide range of chemists especially for specialists in organic and organometallic chemistry. However given work has several serious drawbacks that need to be improved.
From my prospective, the main disadvantage of the manuscript is insufficient analysis of Hiyama coupling application. Authors paid attention for describing different examples of considered reaction of type. However more significant is the comparison of these examples with each other. Authors describe several catalysts for Hiyama reaction. Some of them are based on Palladium (Pd(OAc)2, Palladium chlorides, Pd/C etc). Several approaches are based on applying Copper and Nickel catalysts. How non-experts in this field can choose the best one protocol for their needs? What is the specific advantages of given protocol in comparison with other coupling reaction. Authors should provide more comprehensive analysis. In addition authors should add one or several tables or graphics that summarised analysis of Hiyama reaction.
Authors should significantly change the conclusion part of the work. They should give a brief information about comparison of several reaction protocols.
In my opinion, part devoted to reaction mechanism should be extended. Some brief information about energy barriers, kinetic data should be added.
After performing appropriate corrections, the work can be accepted for publication.
Author Response
We hope that revised manuscript would be satisfying for all requirements and will be suitable for consideration for publication.

Reviewer 2 Report
Dear Editor – in- chief
Hello
Please consider these comments:
1.The authors should explain the novelty of their works.
2. The English language should recheck and edit again.
3. The captions of Figures and Schemes are not correct. Please check them carefully.
4- - Please authors explain that they found the global minima for the drug and Hiyama coupling reaction.
5.In reference part, more papers in recent 5 years should be cited.
Author Response

(The authors gave the same response as above.)

Reviewer 3 Report
Reviewer comments (molecules-1869847):
I assessed the manuscript entitled “Transition metal catalyzed Hiyama cross-coupling: Recent 2 methodology developments and synthetic applications” submitted by Mariusz Mojzych et al. and found that the presented review manuscript is interesting based on a Hiyama cross-coupling reactions.
Some of the important observations were described here:
There are many typographical errors were found within the manuscripts.
--The systematic rearrangements are required based on used catalysts like Pd, Cu and Ni.. .. etc. as a different section and their mechanism.
In some of the schemes, authors used labelling for new catalysts and somewhere not. Therefore, it is advised to use commonly for all used catalysts. Furthermore, I observed that they’re in some of the reactions, in which they mentioned the isomers with their respective yields and their separations also. If such cases are identified with the references, then kindly include them wherever it is applicable.
--There are many missing references based on C-C bond formation reactions using metal catalysts and their various applications.
--There is advised to cite more articles in the introductory part and if possible, then include that discovery also. e.g.
Catal. Sci. Technol., 2019, 9, 5233-5255 DOI: 10.1039/C9CY01331H;
ACS Catal. 2017, 7, 1, 631–651 ACS Catal. 2017, 7, 1, 631–651
J. Org. Chem. 2013, 78, 10, 5022–5025 https://doi.org/10.1021/jo302791q
Chem. Rev. 2015, 115, 17, 9587–9652 https://doi.org/10.1021/acs.chemrev.5b00162
---In the title of this review manuscript, the author used the words “synthetic applications” but I didn’t find the synthetic applications of mentioned schemes.
-- The authors should go for careful proofreading to eliminate (a) grammatical errors; (b) many typos; and also (c) remove unnecessary information and description.
The authors reported some good discoveries based on Transition metal-catalyzed Hiyama cross-coupling reactions but it requires major changes. In my opinion, this manuscript can’t be accepted in its current form but it may be accepted after major revisions.
Author Response

(The authors gave the same response as above.)

Round 2
Reviewer 1 Report
After all improvements from the author side manuscript is suitable for publication.
Reviewer 3 Report
Manuscript ID: molecules-1869847 - Revised Review
After re-assessment of the revised manuscript, I found that this manuscript is now well updated and suitable for publication. Because the authors satisfy all reviewer's comments. In my opinion, it should be accepted in its current form if it fits the journal's requirements.